# Quantum Information Entropy for a Hyperbolic Double Well Potential in the Fractional Schrödinger Equation

**DOI:** 10.3390/e25070988

**Published:** 2023-06-28

**Authors:** R. Santana-Carrillo, J. M. Velázquez Peto, Guo-Hua Sun, Shi-Hai Dong

**Affiliations:** 1Centro de Investigación en Computación, Instituto Politécnico Nacional, UPALM, Mexico City 07700, Mexico; 2ESIME-Culhuacan, Instituto Politécnico Nacional, Av. Santa Ana 1000, Mexico City 04430, Mexico; 3Research Center for Quantum Physics, Huzhou University, Huzhou 313000, China

**Keywords:** hyperbolic double well potential, fractional Schrödinger equation, Shannon entropy, Fisher entropy

## Abstract

In this study, we investigate the position and momentum Shannon entropy, denoted as $S_{x}$ and $S_{p}$, respectively, in the context of the fractional Schrödinger equation (FSE) for a hyperbolic double well potential (HDWP). We explore various values of the fractional derivative represented by *k* in our analysis. Our findings reveal intriguing behavior concerning the localization properties of the position entropy density, $\rho_{s}\left(x\right)$, and the momentum entropy density, $\rho_{s}\left(p\right)$, for low-lying states. Specifically, as the fractional derivative *k* decreases, $\rho_{s}\left(x\right)$ becomes more localized, whereas $\rho_{s}\left(p\right)$ becomes more delocalized. Moreover, we observe that as the derivative *k* decreases, the position entropy $S_{x}$ decreases, while the momentum entropy $S_{p}$ increases. In particular, the sum of these entropies consistently increases with decreasing fractional derivative *k*. It is noteworthy that, despite the increase in position Shannon entropy $S_{x}$ and the decrease in momentum Shannon entropy $S_{p}$ with an increase in the depth *u* of the HDWP, the Beckner–Bialynicki-Birula–Mycielski (BBM) inequality relation remains satisfied. Furthermore, we examine the Fisher entropy and its dependence on the depth *u* of the HDWP and the fractional derivative *k*. Our results indicate that the Fisher entropy increases as the depth *u* of the HDWP is increased and the fractional derivative *k* is decreased.

## 1. Introduction

In recent years, Shannon entropy has garnered significant attention among many researchers in quantum physics, primarily due to its extensive utilization in modern quantum communication systems [1,2,3,4,5,6,7,8,9,10,11,12,13,14,15,16,17,18,19,20,21,22,23,24,25,26,27,28,29,30,31]. These investigations span various branches of physics, such as molecular physics, nuclear physics, atomic physics, and others. The significance of Shannon entropy lies in its role as a generalized version of Heisenberg’s uncertainty principle, offering a quantification of uncertainty in quantum systems and characterizing other physical properties. Of particular interest is the ability of the Shannon entropy to elucidate the localization and delocalization behaviors of particles moving within a confined quantum system. However, it is worth noting that the majority of research on Shannon information entropy has focused on the standard Schrödinger equation, with only a few recent studies delving into its application in the fractional Schrödinger equation [32,33]. The limited exploration of the fractional Schrödinger equation is primarily due to the challenges associated with numerically calculating its solutions.

As we know, the fractional derivative *k* appearing in the kinetic energy operator $\partial^{k}/\partial {\left|x\right|}^{k}$ is taken to be equal to 2 for the usual Schrödinger equation. The FSE emerges as a fundamental framework in fractional quantum mechanics, replacing the conventional kinetic energy operator, $\partial^{2}/\partial x^{2}$, with a fractional derivative indexed by *k* [34]. Initially, the FSE was introduced as a quantum-mechanical model to investigate particle motion governed by Lévy flights, employing the Feynman-integral formalism [34]. Although almost all contributions to this have been developed along with the traditional Schrödinger equation, the FSE exhibits intriguing quantum phenomena due to its fractional derivative index *k* [35,36,37,38,39,40,41,42,43,44,45,46]. These investigations cover a wide range of topics, including energy band structures, light beam propagation dynamics, position-dependent mass FSE, the nuclear dynamics of molecular ion ${H_{2}}^{+}$, Rabi oscillations, spatial soliton propagation, fractional harmonic oscillators, and others. Recent experimental achievements in implementing the FSE in the temporal domain have further bolstered our confidence in exploring this field [47]. These developments undoubtedly enhance our understanding and provide valuable insights into the study of fractional quantum mechanics.

To date, extensive research has been performed on the Shannon entropy in various solvable quantum systems [1,2,3,4,5,6,7,8,9,10,11,12,13,14,15,16,17,18,19,20,21,22,23,24,25,26,27,28,29,30,33]. Notably, hyperbolic soluble potentials hold particular significance in semiconductor physics [18,48,49,50,51]. In a recent study, we investigated the quantum information entropies within the framework of the fractional Schrödinger equation (FSE) for hyperbolic *single* quantum well potential systems [33], considering fractional derivative values within the range k∈(0,2]. At that time, because numerical calculations of results were affected by the presence of wave functions with parity symmetry, we were unable to study the Shannon entropy for the HDWP case. The motivation in studying the double well problem stems from its importance as a toy model in both heterostructure physics and Bose–Einstein condensates [52,53]. Furthermore, its importance has also been demonstrated in other areas of research, such as the Bose–Hubbard model [54,55] and nonlinear Schrödinger equation problems [56,57]. For instance, Lingua and coauthors [54,55] analyzed Shannon-like entropy indicators in a double well system, considering both the position and momentum bases within the framework of the Bose–Hubbard model. They also evaluated the degree of localization and mixing of the ground state in a more complex three-well potential. Similarly, Zhao and his coauthors [56,57] investigated the Shannon entropy for the ground state in nonlinear Schrödinger equation problems.

In this work, our focus is specifically on the study of the HDWP, which is defined as described in Ref. [58]. By exploring this particular potential, we contribute to the understanding of its unique characteristics and shed light on its behavior within the context of the fractional Schrödinger equation. The potential that we study has the form
(1)$$U\left(x\right)=\frac{{ћ}^{2}}{2M}\left(-\frac{u\sinh^{2}\left(x\right)}{\cosh^{4}\left(x\right)}\right).$$
This HDWP, as depicted in Figure 1, exhibits a maximum depth of the potential well equal to u/4 (scaled by the unit $2M/{ћ}^{2}$). It is important to note that the form of the HDWP utilized in this work (see Equation (Equation 1)) differs from our previous study [16], where the HDWP was represented as $-U_{0}\sinh^{4}\left(x\right)/\cosh^{6}\left(x\right)$. In our previous study [16], the Bethe ansatz method was employed, resulting in solutions that are only quasi-exactly solvable.

In order to fill the gap left by the study of a single hyperbolic well in Ref. [33], our current research focuses on investigating the Shannon entropy associated with this double well potential. By examining the global characteristics captured by the Shannon entropy, we aim to compare and contrast them with the local characteristics of the system described by the Fisher entropy. This analysis will provide a comprehensive understanding of the behavior and properties of the double well potential system.

The remainder of this work is structured as follows. Section 2 introduces a fundamental formalism for the solution of the fractional Schrödinger equation (FSE) associated with the hyperbolic potential U(x). In Section 3, we present the results obtained from our analysis. These results encompass the wave functions, the entropy densities $\rho_{s}\left(x\right)$ and $\rho_{s}\left(p\right)$, and the Shannon entropies $S_{x}$ and $S_{p}$ for both the low-lying states and the 10th excited state. We also verify the Beckner–Bialynicki-Birula–Mycielski (BBM) inequality relation. Furthermore, we examine the behavior of the Fisher entropy $F_{x}$ as the derivative *k* and the depth *u* of the HDWP vary. Finally, in Section 4, we summarize our findings and draw our conclusions.

## 2. Formalism

The dimensional FSE is defined as
(2)$$\left(-\frac{{ћ}^{2}}{2M}\frac{\partial^{k}}{{\partial |x|}^{k}}+U\left(x\right)\right)\varphi \left(x\right)=E\varphi \left(x\right),$$
where the fractional derivative *k* is usually taken as the value k∈(0,2] and the eigenvalues and the eigenfunctions are represented by *E* and $\varphi \left(x\right)$, respectively. For numerical solutions of the equation, the fractional derivative can be defined as a second derivative of the wave function φ(x) with respect to a definite integral, incorporating a weighted factor of ${\left|x-\xi \right|}^{1-k}$.
(3)$$\frac{\partial^{k}\varphi \left(x\right)}{{\partial |x|}^{k}}=C_{k}\frac{d^{2}}{dx^{2}}\int_{-\infty }^{\infty }{\left|x-\xi \right|}^{1-k}\varphi \left(\xi \right)d\xi ,$$
where
(4)$$C_{k}=\frac{1}{2\cos\left(\frac{k\pi }{2}\right)\Gamma \left(2-k\right)},$$
which implies that the k≤2.

Now, let us show this numerical method in detail. We first define a factor related to the fractional derivative *k*
(5)$$g_{m}=\frac{{\left(-1\right)}^{m}\Gamma \left(k+1\right)}{\Gamma \left(\frac{k}{2}-m+1\right)\Gamma \left(\frac{k}{2}+m+1\right)},$$
where m=0,1,2,...k>0 and the Γ(x) denotes the Gamma function. This factor $g_{m}$ has the following properties:(6)$$g_{0}\geq 0,\;\;g_{-m}=g_{m}\leq 0,\;\;\left|m\right|\geq 1.$$
On the other hand, the fractional centered difference can be defined as
(7)$$\Delta_{h}^{k}f\left(x\right)=\sum_{m=-\infty }^{\infty }g_{m}f\left(x-mh\right).$$
As a result, one has
(8)$$-\frac{1}{h^{k}}\Delta_{h}^{k}f\left(x\right)=\frac{\partial^{k}}{{\partial |x|}^{k}}f\left(x\right)+O\left(h^{2}\right).$$
When *h* moves to 0, $\frac{\partial^{k}}{\partial |x{|}^{k}}f\left(x\right)$ can be transformed into a fractional derivative with respect to ${\left|x\right|}^{k}$ (k∈(0,2]). Thus, we can rewrite Equation (Equation 2) in matrix form:(9)$$\sum_{l=0}^{N}A_{il}\varphi_{l}=E\varphi_{l}.$$
The eigenvalue problem associated with the fractional Schrödinger equation can be diagonalized, as demonstrated in Ref. [59]. This diagonalization technique allows us to examine the normalized wave functions as a function of the fractional derivative *k*, as illustrated in Figure 2. Notably, for the low-lying states, the wave functions exhibit definite parity. When the value of *k* decreases, the wave functions become more localized towards the double potential wells, and their peaks become more prominent.

Furthermore, an intriguing observation is that the wave functions of the ground state and the first excited state overlap within the rightmost well, while the wave functions of the second and third excited states overlap within the leftmost well. This spatial overlap of the wave functions within each well adds to the complexity and richness of the system’s behavior.

To study the Shannon entropy, we have to calculate the position and momentum entropy densities $\rho_{s}\left(x\right)$ and $\rho_{s}\left(p\right)$, which are defined by [12]
(10)$$\begin{array}{c} \rho_{s}\left(x\right)={\left|\psi \left(x\right)\right|}^{2}\ln\left(|\psi \left(x\right){|}^{2}\right),\\ \rho_{s}\left(p\right)={\left|\phi \left(p\right)\right|}^{2}\ln\left(|\phi \left(p\right){|}^{2}\right). \end{array}$$

Generally, the momentum wave function ϕ(p) can be obtained by the Fourier transformation of the wave function ψ(x),
(11)$$\phi \left(p\right)=\frac{1}{\sqrt{2\pi }}\int \psi \left(x\right)e^{-ipx}dx.$$
In the present study, however, we employ the Fast Fourier algorithm to numerically calculate ϕ(p). Although the wave function can be analytically expressed using the confluent Heun function [58], unfortunately, we do not have access to the Fourier transformation of this function. Therefore, the Fast Fourier algorithm [59] provides an efficient numerical approach to compute the Fourier transform of the wave function, allowing us to analyze the momentum space characteristics of the system.

According to Equation (Equation 10), the position Shannon information entropy $S_{x}$ and the momentum Shannon information entropy $S_{p}$ can be calculated by
(12)$$S_{x}=-\int_{-\infty }^{\infty }\rho_{s}\left(x\right)dx,\;\;\;S_{p}=-\int_{-\infty }^{\infty }\rho_{s}\left(p\right)dp,$$
from which Beckner et al. have obtained an important inequality relation [60,61]
(13)$$S_{x}+S_{p}\geq D\left(1+\ln\pi \right),$$
where *D* denotes the spatial dimension. In this work, we take D=1. This uncertainty relation implies that one of either $S_{x}$ or $S_{p}$ increases but the other will decrease, and vice versa. This relation always remains invariant.

Before concluding this section, it is important to mention the Fisher information, which provides a measure of the local characteristics of quantum systems [62,63], in addition to the Shannon entropy. The Fisher information is defined as [64]
(14)$$I_{F}=\int_{a}^{b}\frac{{\left[\rho^{\prime }\left(x\right)\right]}^{2}}{\rho (x)}dx=4\int_{a}^{b}{\left[\psi^{\prime }\left(x\right)\right]}^{2}dx,$$
where $\rho \left(x\right)={\left|\psi \left(x\right)\right|}^{2}$ denotes the probability density of the wave function.

## 3. Results and Discussion

In this section, we present the results obtained in this study. As shown earlier, the wave functions for the low-lying states are displayed in Figure 2. Utilizing these wave functions, we examine the position and momentum entropy densities, $\rho_{s}\left(x\right)$ and $\rho_{s}\left(p\right)$ (see Figure 3 and Figure 4), as well as their corresponding Shannon entropies, $S_{x}$ and $S_{p}$. To investigate the behavior of the position and momentum entropy densities for higher excited states, we also analyze the 10th excited state (see Figure 5 and Figure 6).

We observe that the position entropy density, $\rho_{s}\left(x\right)$, for the ground state and the first excited state is nearly identical, as depicted in Figure 4. This behavior can be explained by referring to the wave function plots in Figure 2. As the derivative *k* becomes very small, their differences gradually become apparent. However, it is important to note that these slight differences primarily arise from the numerical calculations.

Furthermore, we find that the position entropy density, $\rho_{s}\left(x\right)$, for higher excited states, such as the 10th excited state, becomes more localized as the derivative *k* decreases. Conversely, the momentum entropy density, $\rho_{s}\left(p\right)$, exhibits the opposite trend, becoming more delocalized as the derivative *k* decreases.

The Shannon entropies, $S_{x}$ and $S_{p}$, calculated using Equation (Equation 11), are illustrated in Figure 7 and Figure 8, respectively. It is evident that $S_{x}$ increases with increasing *k* for a given parameter *u*, while $S_{p}$ decreases. Additionally, as the depth *u* of the potential well increases, $S_{x}$ decreases, whereas $S_{p}$ increases. This behavior can be attributed to the increased confinement of the particle within the respective well as the potential well becomes deeper, resulting in greater stability.

Importantly, it is worth noting that the sum of the Shannon entropies, $S_{x}$ and $S_{p}$, still satisfies the Beckner–Bialynicki-Birula–Mycielski (BBM) inequality given in Equation (Equation 12), as demonstrated in Figure 9.

Finally, we investigate the Fisher entropy, $I_{F}$, as shown in Figure 10, and observe that it increases with an increase in the depth *u* of the double well potential. Conversely, the Fisher entropy decreases as the derivative *k* increases. This behavior indicates that as the potential well becomes deeper, the local characteristic of the system becomes more prominent, leading to an increase in the Fisher entropy. Conversely, as the derivative *k* increases, the system exhibits a reduced local characteristic, resulting in a decrease in the Fisher entropy.

## 4. Concluding Remarks

In this work, we have investigated the Shannon entropy $S_{x}$ and $S_{p}$ for a hyperbolic double well potential using the time-independent fractional Schrödinger equation. We have examined the variations of the wave function, entropy density $\rho_{x}$ and $\rho_{p}$, and Shannon entropy $S_{x}$ and $S_{p}$ with respect to the fractional derivative *k*. Additionally, we have verified the satisfaction of the BBM inequality relation and illustrated the behavior of these quantities as the depth *u* of the double well increases. Notably, due to the definite parity of the wave function, these physical quantities exhibit symmetric properties around the point x=0.

We have observed that the wave functions for the ground state and the first excited state overlap in the rightmost well, while those of the second and third excited states overlap in the leftmost well. Furthermore, we have found that as the depth *u* of the potential well increases, the particle becomes more confined within the respective well, resulting in increased stability. Moreover, decreasing the depth *u* of the HDWP and the fractional derivative *k* also contributes to increased particle stability.

Finally, it is worth mentioning that the formalism presented in this work can be widely applied to various quantum systems, including unsolvable ones, due to the numerical approach employed in our study.

## Figures and Tables

**Figure 1 entropy-25-00988-f001:**
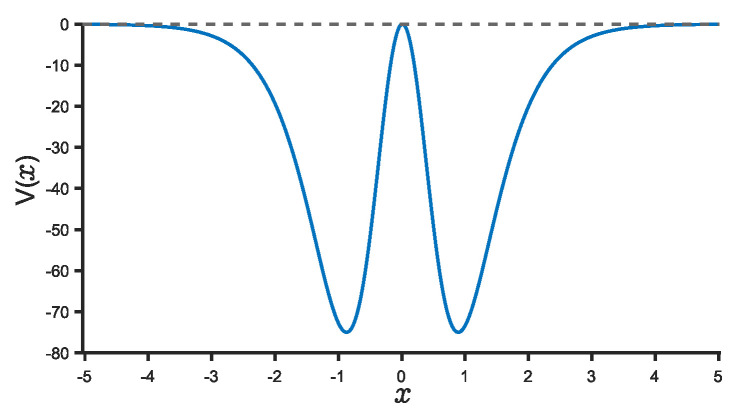
(Color online) Plot of the HDWP given in (1) V(x) ($2M\;U\left(x\right)/{ћ}^{2}$) with respect to *x*. Its maximum depth is u/4 (u=300).

**Figure 2 entropy-25-00988-f002:**
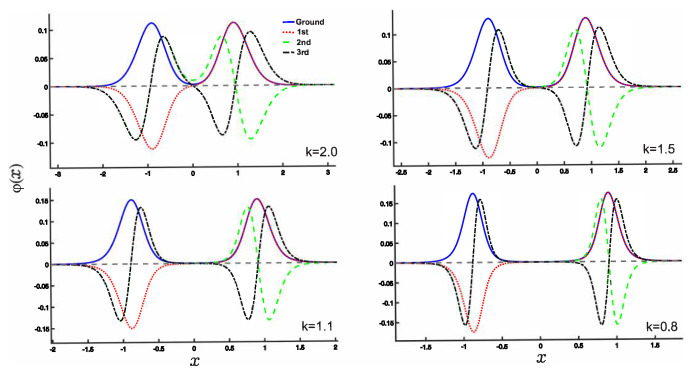
(Color online) Plots of the normalized wave functions for the HDWP (1). The fractional derivative *k* is taken as the values 2.0,1.5,1.1,0.8. The solid blue line, the red dotted line, the green dashed line, and the black dash-dotted line denote the ground state and the 1st, 2nd, and 3rd excited states, respectively. Here, we take ћ=2M=1.

**Figure 3 entropy-25-00988-f003:**
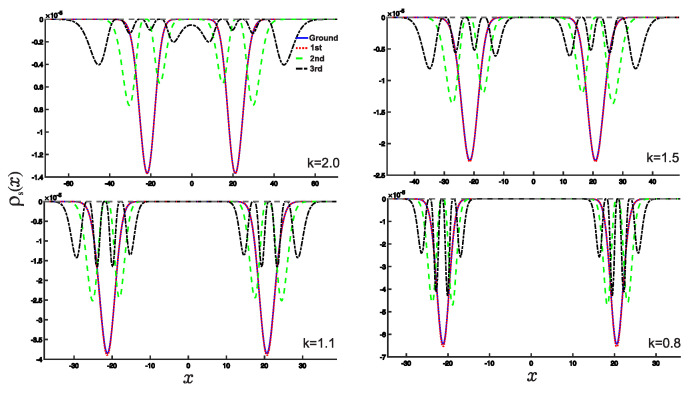
(Color online) Plots of $\rho_{s}\left(x\right)$ as a function of the variable *x* for different values of the *k*. *k* is taken as the values 2.0,1.5,1.1,0.8. The notations of the different lines are the same as in Figure 2.

**Figure 4 entropy-25-00988-f004:**
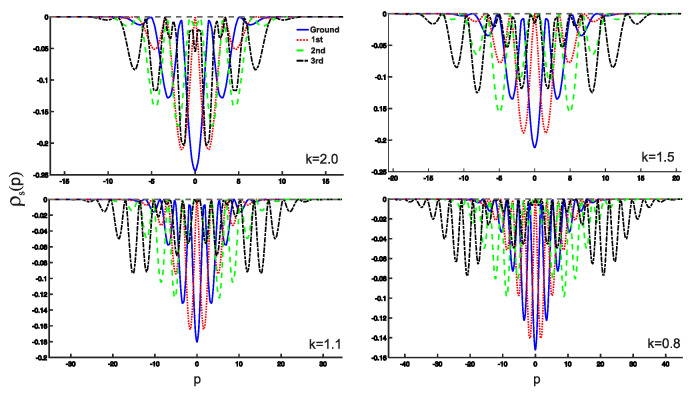
(Color online) Plots of momentum entropy density $\rho_{s}\left(p\right)$ as a function of the variable *p*.

**Figure 5 entropy-25-00988-f005:**
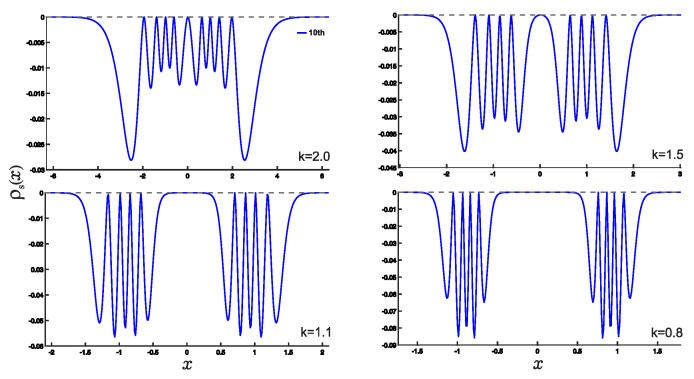
(Color online) Position entropy density $\rho_{s}\left(x\right)$ plots as a function of the position *x* for normalized 10th excited state. The variable *k* takes the values of 2,1.5,1.1,0.8 for the fractional derivative.

**Figure 6 entropy-25-00988-f006:**
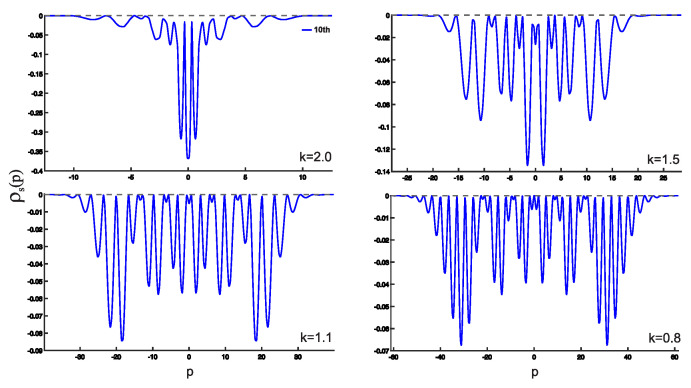
(Color online) Same as Figure 5 but for the momentum entropy density $\rho_{s}\left(p\right)$ for the 10th excited state.

**Figure 7 entropy-25-00988-f007:**
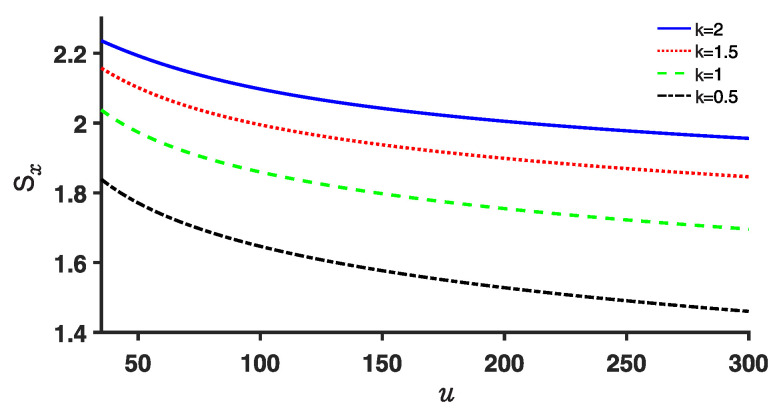
(Color online) Plot of position entropy $S_{x}$ for the ground state. The values of the fractional derivative *k* are taken as above.

**Figure 8 entropy-25-00988-f008:**
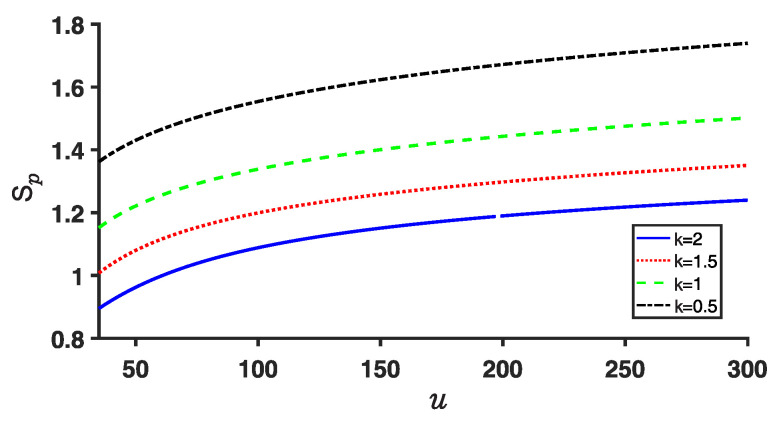
(Color online) Same as Figure 7 but for the plots of momentum entropy $S_{p}$ for the ground state.

**Figure 9 entropy-25-00988-f009:**
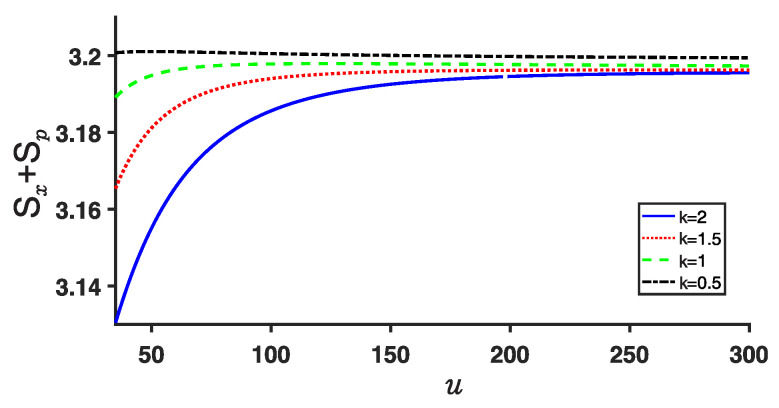
(Color online) Same as Figure 7 and Figure 8 but for their sum $S_{x}+S_{p}$ for the ground state.

**Figure 10 entropy-25-00988-f010:**
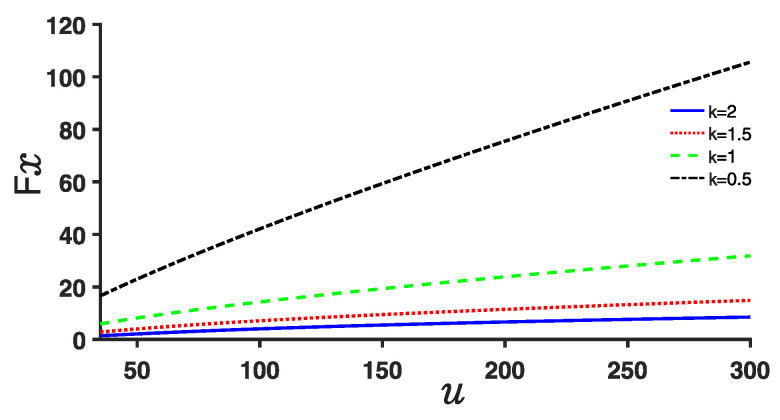
(Color online) Plot of the Fisher entropy for different values of the depth parameter *u* and fractional derivative *k*.

## Data Availability

Not applicable.

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
