# Peer review of "Quantum Information Entropy for a Hyperbolic Double Well Potential in the Fractional Schrödinger Equation"

_entropy, 2023, doi:10.3390/e25070988_

Round 1
Reviewer 1 Report
Please see the attached pdf file.

English should be improved.
Reviewer 2 Report
This manuscript presents considers the low energy eigenstates of a double well potential in the fractional Schrödinger equation and presents calculations of the quantum information entropy for these states.
The topic of research is very specific and not very well motivated. The paper is mostly an exercise in calculation following a methodology that the authors applied previously to the case of a single well potential. The findings are merely described, but not interpreted. In addition, the manuscript is weirdly structured (in the introduction it would make more sense for the authors to start with the FSE and the motivations for studying it, then define and discuss the Shannon entrory).
All in all, this is incremental work which does not add much knowledge to the literature. I cannot recommend it for publication in Entropy.
The quality of the English is reasonable.
Author Response
see attached respond.

Reviewer 3 Report
The authors presented a study focused on the analysis of the quantum information entropy for a hyperbolic double well potential within the fractional Schrödinger equation formalism. This is a very interesting work, whose conclusions are well-supported by the results. Therefore, I think that this paper should be accepted for publication in Entropy. I only suggest the authors can briefly discuss the potentialities of this formalism in studying similar model systems.
Author Response
Thanks for your good suggestions. In the final Section IV, we have added one paragraph to address this potentiality.
Round 2
Reviewer 1 Report
I think that the authors have fully addressed the points which I raised and I now recommend the article for publication.
I don't feel qualified to assess the quality of English, but some minor typos may be still present.
Reviewer 2 Report
The authors have made (some) efforts clarifying their motivations, but the changes made to the manuscript are minimal, particularly regarding the interpretation and significance of the observed results. My opinion is thus unchanged and I cannot give a positive recommendation for publication in Entropy.
The quality of the English is reasonable.